

# Roads affect the spatial structure of butterfly communities in grassland patches

Piotr Skórka[1], Magdalena Lenda[1,2] and Dawid Moroń[3]

[1] Institute of Nature Conservation, Polish Academy of Sciences, Kraków, Poland
[2] School of Biological Sciences, The University of Queensland, Brisbane, Australia
[3] Institute of Systematics and Evolution of Animals, Polish Academy of Sciences, Kraków, Poland

Corresponding author
Piotr Skórka, skorasp@gmail.com,
pskorka@iop.krakow.pl

## ABSTRACT

Roads may have an important negative effect on animal dispersal rate and mortality and thus the functioning of local populations. However, road verges may be surrogate habitats for invertebrates. This creates a conservation dilemma around the impact of roads on invertebrates. Further, the effect of roads on invertebrates is much less understood than that on vertebrates. We studied the effect of roads on butterflies by surveying abundance, species richness and composition, and mortality in ten grassland patches along high-traffic roads (∼50–100 vehicles per hour) and ten reference grassland patches next to unpaved roads with very little traffic (<1 vehicle per day) in southern Poland. Five 200-m transects parallel to the road were established in every grassland patch: at a road verge, 25 m from the verge, in the patch interior, and 25 m from the boundary between the grassland and field and at the grassland-arable field boundary. Moreover, one 200-m transect located on a road was established to collect roadkilled butterflies. The butterfly species richness but not abundance was slightly higher in grassland patches adjacent to roads than in reference grassland patches. Butterfly species composition in grasslands adjacent to roads differed from that in the reference patches. Proximity of a road increased variability in butterfly abundances within grassland patches. Grassland patches bordering roads had higher butterfly abundance and variation in species composition in some parts of the grassland patch than in other parts. These effects were not found in reference grassland patches, where butterfly species and abundance were more homogenously distributed in a patch. Plant species composition did not explain butterfly species. However, variance partitioning revealed that the presence of a road explained the highest proportion of variation in butterfly species composition, followed by plant species richness and abundance in grassland patches. Road mortality was low, and the number of roadkilled butterflies was less than 5% of that of all live butterflies. Nevertheless, the number and species composition of roadkilled butterflies were well explained by the butterfly communities living in road verges but not by total butterfly community structure in grassland patches. This study is the first to show that butterfly assemblages are altered by roads. These results indicate that: (1) grassland patches located near roads are at least as good habitats for butterflies as reference grassland patches are, (2) roads create a gradient of local environmental conditions that increases variation in the abundance of certain species and perhaps increases total species richness in grassland patches located along roads, and (3) the impact of roads on butterflies is at least partially independent of the effect of

plants on butterflies. Furthermore, (4) the direct impact of road mortality is probably spatially limited to butterflies living in close proximity to roads.

## INTRODUCTION

Roads can exert severe impacts on animal populations (*Benítez-López, Alkemade & Verweij, 2010*; *Matos et al., 2017*), either through direct road mortality (collisions with cars) or through habitat fragmentation and barrier effects increasing the isolation of populations (*Trombulak & Frissell, 2000*; *Forman et al., 2003*; *Tanner & Perry, 2007*; *Schuster, Römer & Germain, 2013*). To date, most studies have focused on the estimates of road mortality, since millions of individuals from a wide range of taxonomic groups are being killed every year (*Coelho, Kindel & Coelho, 2008*; *Grilo, Bissonette & Santos-Reis, 2009*; *Brzeziński, Eliava & Żmihorski, 2012*). However, roads also change the nearby environment by increasing influxes of salt and pollutants and changing the microclimate or water regime (*Forman, 2000*; *Forman et al., 2003*; *Jackson & Jobàggy, 2005*; *Green, Machin & Cresser, 2006*). These changes are especially well exhibited in the changes in plant communities near roads compared with changes in more distant habitats (*Lee, Davies & Power, 2012*; *Neher, Asmussen & Lovell, 2013*). Many herbivores are dependent on specific plants (*Stam et al., 2013*). Among insects, grassland butterflies are directly dependent on several plant species during different parts of their life cycle. Butterfly larvae usually have specific host-plant species/groups, while adults often use specific plant species as nectar sources (*Munguira, Garcia-Barros & Cano, 2009*). As a result, a strong correlation between butterfly and plant species richness is often observed in grassland patches (*Skórka, Settele & Woyciechowski, 2007*; *Chmura, Adamski & Denisiuk, 2013*). It may thus be expected that roads decrease the species richness and abundance of butterflies and plants and alter the species composition of the community in such a way that some species may tolerate adjacency while others avoid it. Moreover, alterations to plant communities, especially to host plants and nectar sources, by the creation or existence of roads should also be indirectly related to grassland butterfly community structure, species richness and abundance. However, roads may also change insect communities in a different manner. Road verges, which are linear grassy structures accompanying roads, are regarded as a good surrogate habitat for plants and may act as their dispersal corridors (*Tikka, Högmander & Koski, 2001*; *Kalwij, Milton & McGeoch, 2008*). This potential positive effect may, of course, be diminished by road mortality, which can be high in butterflies (*Skórka et al., 2015*; *Baxter-Gilbert et al., 2015*). However, it is unknown whether road mortality affects all butterflies in a grassland patch adjacent to a road or is confined to the part of the grassland patch located directly along road (e.g., the road verge). Therefore, the complexity of the possible direct and indirect effects that roads can exert on butterflies make these insects especially interesting subjects to study in order to understand the impacts of roads on insects at the community level. This is essential

for effective mitigation measures to minimize the impacts of existing and future roads on insect populations, a subject that is rarely studied despite the important ecological roles that insects play (*Baxter-Gilbert et al., 2015*; *Munoz, Torres & Gonzalez-Megias, 2015*).

In this paper, we analyzed the effect of road proximity on plant and butterfly communities, and we tested the following hypotheses (also see Table 1):

1. Grassland patches adjacent to roads with traffic have lower butterfly and plant species richness and abundance/cover than grassland patches located far from roads

2. Grassland patches adjacent to roads with traffic have different butterfly and plant species compositions than grassland patches located far from roads

3. Closer adjacency of roads increases variability in butterfly and plant richness, abundance and species composition within grassland patches

4. Roads impose direct and indirect (via plants) effects on butterfly species composition in grassland patches

5. Road mortality is explained by the abundance of all butterflies in the entire grassland patch adjacent to the road

## STUDY AREA AND METHODS

### Study areas

The study was conducted in grasslands in the vicinity of Kraków, Proszowice and Tarnów (southern Poland). Grasslands are major habitats for European butterflies (*Settele et al., 2009*). We selected 10 grassland patches adjacent to roads with heavy traffic (national and provincial roads with ∼50–100 vehicles per hour and one lane in each direction) and 10 reference (control) grassland patches located far (at least 200 m) from major roads, with access via an unpaved field road (<1 vehicle per day, Fig. S1). The traffic on the studied paved roads was average for the traffic volume on Polish roads (*Opoczynski, 2016*). Such roads constitute approximately 30% (∼120,000 km) of the length of all roads in Poland (*Opoczynski, 2016*).

The grassland patches were similar in size (7.8–13.5 hectares) and type (wet grassland associated with *Molinietalia* type) The dominant plant species for this grassland type are the grasses *Festuca pratensis, Deschampsia cespitosa,* and *Elymus repens* and numerous flowering plants such as *Cirsium arvense, Sanguisorba officinalis, Taraxacum officinale, Centaurea jacea,* and *Lythrum salicaria* (File S2).

We ensured that the two grassland types did not differ in their covers of arable fields ($t$-test: $t = 0.578$, $df = 17.98$, $P = 0.571$; mean $\pm$ *SE*: 62.6 $\pm$ 2.6%, range: 42–80%), woodland ($t$-test, $t = -0.499$, $df = 17.94$, $P = 0.624$; mean $\pm$ *SE*: 7.3 $\pm$0.8%, range: 2–12%), human settlements ($t$-test, $t = -0.052$, $df = 17.60$, $P = 0.958$; mean $\pm$ *SE*: 8.6 $\pm$ 0.9%, range: 3–15%) and other grasslands ($t$-test, $t = -0.451$, $df = 17.99$, $P = 0.657$; mean $\pm$ *SE*: 21.4 $\pm$ 2.4%, range: 6–42%) in a 500 m buffer around the boundary of each habitat patch. The studied grasslands were mown twice per year: once at the beginning of June and once in mid-August. This is a typical mowing scheme of grasslands in Poland.

In every grassland patch near a road, we established 5 transects that were each 200 m long, along which butterflies and plants were surveyed (Fig. S1). The first transect was

Skórka et al. (2018), *PeerJ*, DOI 10.7717/peerj.5413

**Table 1** Summary of major results and support for hypotheses.

| | Hypothesis | Explanations/Predictions | Statistical test | Support | Comments |
|---|---|---|---|---|---|
| 1. | Grassland patches adjacent to roads with traffic have lower butterfly and plant species richness and abundance/cover than grassland patches located far from roads. | Influxes of pollutants, salt and road mortality potentially negatively affect animal and plant populations. Thus, presence of roads should have negative effects on diversity indices. | Generalized linear models with negative binomial error variance. | No | Butterfly species richness was higher in grassland patches adjacent to roads than in grassland patches located far from roads. No differences were found in butterfly abundance nor in plant species richness and cover between two types of grassland. |
| 2. | Grassland patches adjacent to roads with traffic have different butterfly and plant species compositions than grassland patches located far from roads. | Roads change conditions in adjacent grassland patches. Thus it is expected that species composition and species abundances are different (but without specifying the difference) in grassland patches adjacent to roads than in grassland patches located far from roads. | Partial redundancy analysis (butterflies), partial canonical correspondence analysis (plants). | Yes/No | Species composition of butterflies (but not plants) was different in grassland patches adjacent to roads than in grassland patches located far from roads. |
| 3. | Closer adjacency of roads increases variability in butterfly and plant richness, abundance and species composition within grassland patches. | Roads change environmental conditions in areas adjacent to roads. Thus, this should create gradient of conditions from the road verge towards habitat patch interior. Hence, species richness, abundance and composition should vary in different parts of a grassland patch adjacent to road. This should not be visible in grassland patches located far from roads. | Generalized linear models with negative binomial error variance (species richness and abundance/cover), partial redundancy analysis (species composition of butterflies), partial canonical correspondence analysis (plants). | Yes/No | Butterfly and plant abundance/cover and species composition (but not species richness) differed among transects located in different parts of a grassland patch adjacent to road. This was not found in grassland patches located far from roads. |
| 4. | Roads impose direct and indirect (via plants) effects on butterfly species composition in grassland patches. | Roads change conditions in adjacent grassland patches that may affect both butterflies and plants. However, butterflies are herbivores strongly dependent on plants. Thus, species richness, abundance and composition of butterflies in grassland patches may be affected directly by roads (e.g., road mortality) and indirectly by plants (e.g., species composition, cover). | Co-correspondence analysis, correlation analysis, hierarchical variance partitioning (separating direct effect of road adjacency on butterfly species composition from the effect of plant species on butterfly species composition). | Yes/No | Plant species composition did not explain butterfly species composition in grassland patches. However, butterfly species richness correlated with plant species richness. In total both adjacency of a road and plant species richness and abundance had significant individual impact on butterfly species composition in grassland patches. |

Skórka et al. (2018), *PeerJ*, DOI 10.7717/peerj.5413

**Table 1** (*continued*)

| | Hypothesis | Explanations/Predictions | Statistical test | Support | Comments |
|---|---|---|---|---|---|
| 5 | Road mortality is explained by the abundance of all butterflies in the entire grassland patch adjacent to the road. | Road mortality is one of the most direct effects of roads on butterflies. However, road mortality may affect (1) entire species population in a grassland patch adjacent to road or (2) only part of the population occurring near the road (e.g., on road verge). Thus, if the first is true the species richness, abundance and composition of live butterflies in a grassland patch explain the composition and number of roadkills. If road have spatially limited impact then data on butterflies on road verges explain composition and number of roadkills. | Correlation analysis, co-correspondence analysis. | No | Butterfly species richness and composition on road verges rather than in entire grassland patch better explained number and species composition of roadkilled butterflies. |

located at the verge (the border between the grassland and the road but not including asphalt). The second and third transects were located 25 m from the edge of the road and in the patch interior. The fourth transect ran 25 m from the border between the grassland patch and arable field. The fifth transect ran along the border between the grassland and cropland. The purpose of this design was to test hypotheses 1–3 and to separate the effect of the road from the impact of the border itself. Changes in habitat conditions (soil and vegetation) occur in most areas to a distance of approximately 50 m from the edge of the road (*Forman & Alexander, 1998*).

The transects in reference grassland patches were identical to those in grassland patches adjacent to roads with traffic. In all reference grassland patches, there were unpaved field roads that allowed farmers to reach the area. However, these roads were grassy, and the number of vehicles was less than one per day.

## Butterfly and plant surveys

Butterflies were counted along transects during twelve surveys from mid-April to mid-September in approximately 10–14 day intervals in 2013. European butterflies have different developmental modes; some species are multivoltine (e.g., *Araschnia levana*), but there are also species that only occur during a short period in a season (e.g., *Callophrys rubi*) (*Settele et al., 2009*). Consecutive transect counts were established to cover the entire flying period of butterfly species with different biologies. The observer traveled the centerline of a 5 m wide transect. Counts along transects (Pollard walks) is the standard and most commonly used method to study populations of butterflies (*Pollard & Yates, 1993*). Observations were carried out during good weather (minimum temperature of 17 °C, wind up to 3° on the Beaufort scale, and cloudy to 25%).

During each visit, we also collected roadkilled butterflies along the 200 m part of the road neighboring the studied grasslands (near both major roads and unpaved roads, but we did not find any dead butterflies at the latter). These transects were adjacent to the transect located at the road verge where living butterflies were counted (Fig. S1).

Plants were surveyed in one 4 × 10 m rectangular plot located in the middle of each transect. Surveys were performed at the beginning of July. During the surveys, we noted the coverage of each plant species (*Mueller-Dombois & Ellenberg, 1974*).

Field procedures were approved by the panel experts in the Ministry of Science and Higher Education (approval: 0301/B/P01/2010/39). We received permission from farmers (owners) of the land to conduct field surveys. Collecting butterflies on roads did not require any formal permits because these were public roads.

## Statistical analysis

Hypothesis 1. We used a generalized linear model (GLM) with a negative binomial error structure and log-link function to test differences in butterfly and plant species richness, butterfly abundance and plant cover between grassland patches adjacent to roads and reference grassland patches. In the case of butterflies, we calculated the sum of the individuals from five transects located within a grassland patch to obtain proxies of total population sizes and because there was little difference in butterfly abundance

among the five transects within grassland patches (*Rosin et al., 2012*; *Skórka et al., 2013a*). In addition, we also used an individual-based rarefaction technique (*Heck Jr, Van Belle & Simberloff, 1975*) to test the differences in butterfly species number between grassland types while taking sampling effort into account. This analysis calculated the expected number of species based on the number of individuals sampled. We did not perform analyses for plants since we did not count individuals due to the difficulty in defining what an individual plant is.

Hypothesis 2. We used partial RDA to test whether or not butterfly species composition and the abundance of each species differed between grassland patches adjacent to roads with traffic and reference grassland patches. We included the following covariates so that their effects were removed: transect location within a patch and plant richness and cover. Moreover, we performed partial redundancy analysis (partial RDA) using butterfly data summed across transects for every grassland patch. However, the results were similar enough that the conclusions were the same for either analysis. We used this method because the longest ordination axis in a detrended canonical correspondence analysis was short (1.4); thus, the linear method was preferred (*Jongman, Ter Braak & Van Tongeren, 1987*).

In the case of plants, we used partial canonical correspondence analysis (partial CCA) to test whether or not plant species composition and mean cover differ between grassland patches adjacent to roads and reference grassland patches. We designated transect location as a variable to remove its effect. Plant cover was square root transformed before analysis.

We used 1,000 Monte Carlo permutations to test the statistical significance of the ordination axes.

Hypothesis 3. We used a generalized linear mixed model (GLMM) with a negative binomial error structure and log-link function to test differences in butterfly and plant species richness, butterfly abundance and plant cover between transect locations in different parts of the grassland patches. Grassland patch identifier (site number) was assigned as a random effect. Analyses were conducted separately for grassland patches located near roads and reference ones. To determine which levels of the categorical factor (transect location) were significantly different, we used paired contrast analysis.

We used partial RDA to test whether or not butterfly species composition and abundances differ between transect locations in different parts of the grassland patches. Transect location was treated as a categorical variable. We included the following covariates so that their effects were removed: grassland patch identifier and combined plant richness and cover (first principal component—PC1).

We used partial CCA to test whether or not plant species composition and cover differed between transect locations in different parts of the grassland patches. Transect location was treated as a categorical variable. We assigned patch identifier as a supplementary variable to remove its effect. Plant cover was square root transformed before analysis. We used 1,000 Monte Carlo permutations to test the statistical significance of ordination axes and to determine which transect location contributed significantly to the differentiation of plant communities.

The partial RDA and partial CCA were calculated separately for grassland patches adjacent to roads and reference ones.

Hypothesis 4. We performed variance partitioning analysis to test the relative contributions of (1) grassland type (adjacency of road or its absence), (2) plant diversity and (3) their joint effect to the differentiation of butterfly community composition. Plant diversity was represented by the first principal component, which was associated with the number of species and total cover of plants (PC1 explained 81% of variation between the two variables).

We used co-correspondence analysis (Co-CA) (*Braak ter & Schaffers, 2004*) to reveal whether or not butterfly species composition and abundance can be explained by plant community composition. In this analysis, we used butterfly data summed and plant data averaged across the entire grassland patch because Co-CA currently does not allow for the effective removal of variation explained by covariates (*Braak ter & Schaffers, 2004*).

Hypothesis 5. We used Co-CA to test the relationship between the butterfly community living in the grassland patch adjacent to roads and the species composition of roadkilled butterflies. Two Co-CA analyses were performed. First, we related the species composition of roadkilled butterflies to the composition of live butterfly species recorded in all locations within a grassland patch to examine whether or not road mortality is associated with the butterfly community measured across an entire grassland patch. Second, we related the abundance of roadkilled butterflies to the abundances of live species in road verges. Given these two analyses, we could infer whether or not the impact of road mortality is grassland wide or spatially limited to the road verge. In addition to Co-CA, we used correlation analysis to seek a link between the total species richness and abundance of living butterflies and the species richness and abundance of roadkilled ones. Again, two sets of correlation analyses were performed: (1) one for living butterflies summed across the entire grassland patch and (2) one for butterflies living on road verges only.

All GLMM, GLM and correlation analyses were performed in SPSS 23 software (IBM, Armonk, NY, USA). All partial CCA, partial RDA and Co-CA analyses were performed in Canoco 5.0 software.

## RESULTS

Altogether, 14,017 individuals belonging to 42 butterfly species were recorded during the study. A total of 6,922 individuals belonging to 42 butterfly species were recorded in patches adjacent to roads, while 7,095 individuals belonging to 38 species were recorded in reference grassland patches (Table S1). In both grassland types, the dominant species (>5% of all individuals) were *Pieris rapae* (1 635 individuals), *Coenonympha pamphilus* (1 472), *Aphantopus hyperantus* (963), *Gonepteryx rhamni* (949), *Polyommatus icarus* (869), and *Inachis io* (746). Species that occurred only in grassland patches adjacent to roads with traffic were *Callophrys rubi* (57 individuals), *Plebejus argyrognomon* (20), *Nymphalis antiopa* (8), and *Argynnis aglaja* (5).

Hypothesis 1. In total, there were significantly more butterfly species in grassland patches adjacent to roads with traffic than in reference grassland patches (GLM $F_{1,18} = 5.545$, $P = 0.034$, Fig. 1A) after controlling for plant richness and cover (PC1plant; $F_{1,17} = 1.327$, $P = 0.265$). There were no differences in mean butterfly abundance (GLM $F_{1,17} = 0.133$,

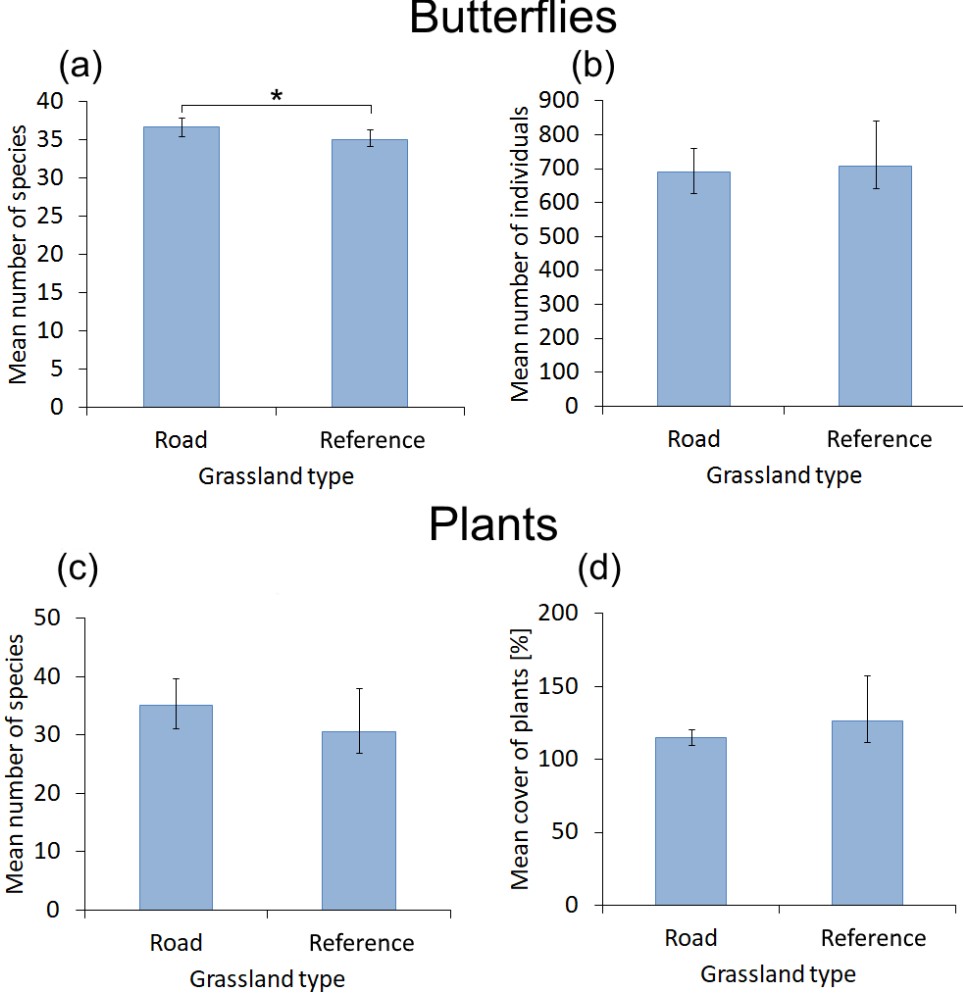

**Figure 1  The impact of grassland location on the number of butterfly species and individuals.** The impact of grassland type on the mean number of butterfly species (A) and individuals (B) and the mean number of plant species (C) and cover (D) within grassland patches adjacent to roads with traffic and those far from roads. Whiskers are 95% confidence intervals. Note: * indicates a statistically significant difference at $P < 0.05$.

$P = 0.720$, Fig. 1B) between the two grassland types, but the abundance was marginally positively correlated with plant species and cover (PC1plant; GLM $F_{1,17} = 4.397$, $P = 0.051$). Additionally, rarefaction analysis revealed that the estimated number of butterfly species was higher in grassland patches located near roads than in reference grasslands (Fig. S2).

There were no differences in total plant species richness (GLM $F_{1,18} = 2.985$, $P = 0.101$, Fig. 1C) nor in mean cover (GLM $F_{1,18} = 2.340$, $P = 0.144$, Fig. 1D) between grassland patches located near roads and those located away from roads.

Hypothesis 2. Partial RDA showed that the butterfly communities significantly differed between grassland patches located near roads with traffic and reference patches ($F = 6.1$,

$P = 0.0005$, Fig. 2). The two grassland types accounted for 15.4% of the variation in species composition (Fig. 2).

The partial CCA revealed that plant communities did not differ ($F = 0.8$, $P = 0.801$) between grassland patches located near roads with traffic and reference ones (Fig. 2).

Hypothesis 3. In grassland patches adjacent to roads with heavy traffic, there were no differences in butterfly species richness among the five transects located at different distances from a road (GLMM, $F_{4,45} = 1.394$, $P = 0.251$, Fig. 3). However, the mean abundance of butterflies varied depending on the location of the transect within the grassland patch (GLMM, $F_{4,45} = 3.440$, $P = 0.015$, Fig. 3). Contrast analysis revealed that abundance was significantly higher in transects located inside grassland patches and 25 m from road verges than in transects at the field boundary (Fig. 3).

Partial RDA showed that the location of a transect explained a significant ($F = 2.0$, $P = 0.0005$) amount (18.4%) of the variation in butterfly species composition within grassland patches adjacent to roads with traffic (Fig. 4). The first ordination axis separated butterfly communities along transects near grassland patch boundaries from butterfly communities along transects located inside the grassland patch (Fig. 4). The second ordination axis mostly separated butterfly communities along transects near field boundaries from the butterfly communities along the transects at road verges (Fig. 4). Accordingly, the tests indicated that transects located inside habitat patches ($F = 3.0$, $P = 0.0005$), at road verges ($F = 2.3$, $P = 0.01$) and at field boundaries ($F = 2.0$, $P = 0.0167$) contributed significantly to the differentiation of butterfly communities within grassland patches adjacent to roads with traffic (Fig. 4).

In the reference grassland patches, there were no differences in butterfly species richness (GLMM, $F_{4,45} = 0.142$, $P = 0.966$, Fig. 3) nor in mean abundance (GLMM $F_{4,45} = 1.476$, $P = 0.225$, Fig. 3) among the five transects located in different parts of the patch.

The first two axes of partial RDA showed that the location of transects within the patches explained 11.7% of the variation in butterfly species composition in reference grasslands (Fig. 4). The ordination axes were statistically significant (test of all axes, $F = 1.5$, $P = 0.0345$). The first ordination axis separated butterfly community transects located 25 m from field boundaries from the other four transects ($F = 2.6$, $P = 0.04$).

In grassland patches adjacent to roads with traffic, there were no differences in plant species richness (GLMM, $F_{4,45} = 2.359$, $P = 0.068$, Fig. S3). However, mean plant species cover varied depending on the location within the patch (GLMM, $F_{4,45} = 3.190$, $P = 0.022$, Fig. S3). Contrast analysis revealed that cover was statistically higher at road verges than in any other part of the grassland patch except the patch interior (Fig. S3).

The first two axes of the partial CCA explained 10.1% of the variation in plant species composition (pseudo $F = 1.4$, $P = 0.0005$), and the location of transects within the patch explained 13.7% of this variation (Fig. S4). The first ordination axis separated plant communities along transects located in road verges and 25 m from the road verge from plant communities along transects located inside the habitat patch (Fig. S4). The second ordination axis separated plant communities along transects located at the field boundary and 25 m from road verges from those recorded along other transects within grassland patches (Fig. S4). However, tests of the contribution of transect locations to the ordination

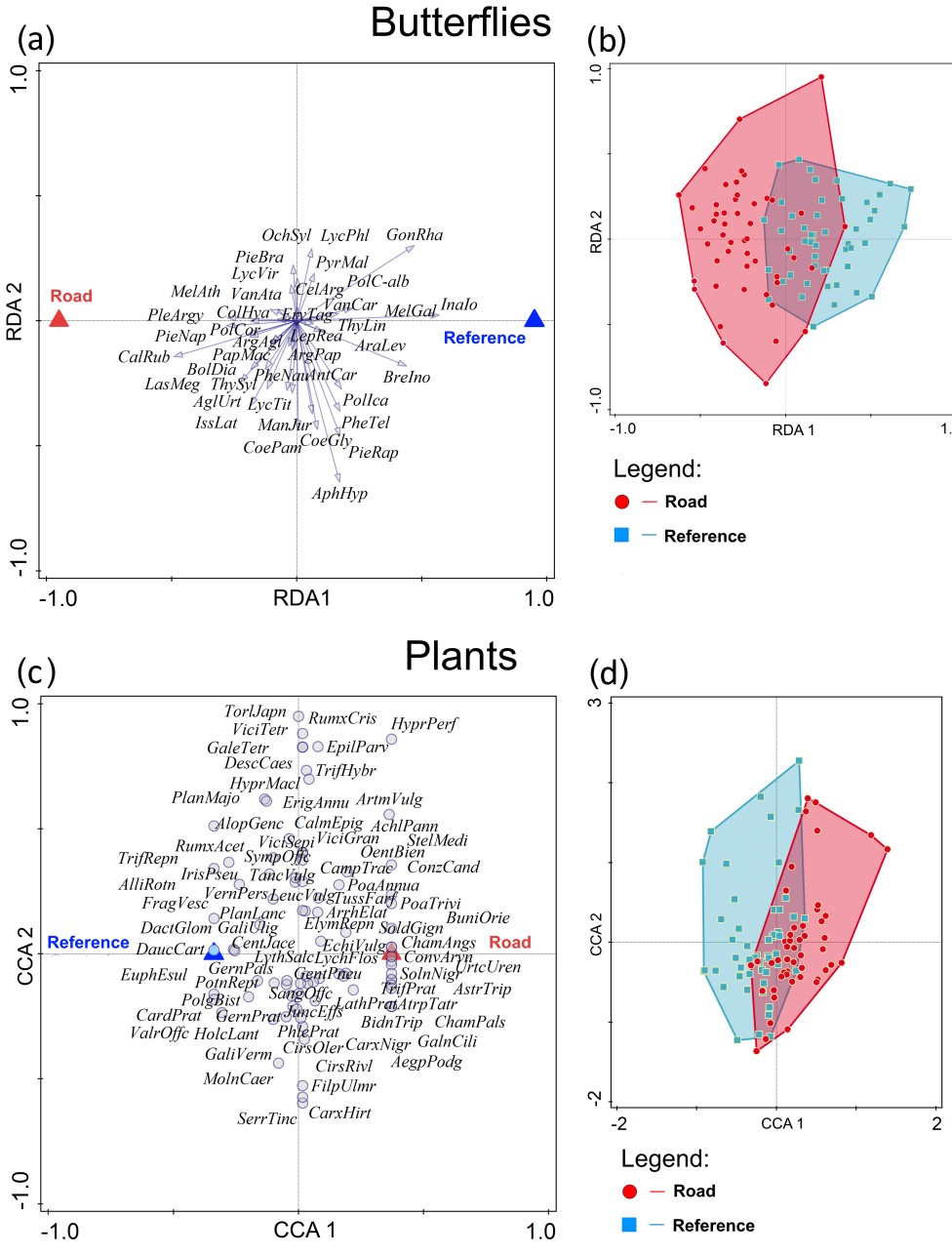

**Figure 2** **Differentiation of butterfly and plant species composition in grassland patches adjacent to roads and those farm from roads.** Ordination of butterfly (A, B) and plant (C, D) species in grassland patches adjacent to roads with traffic and those far from roads. Partial redundancy (butterflies) and partial canonical correspondence (plants) analyses were used for ordination of species after removing the effects of transect location within a patch. Road, grassland patches adjacent to roads; Reference, grassland patches located far from roads. Species abbreviations are the first letters of the genus and species names.

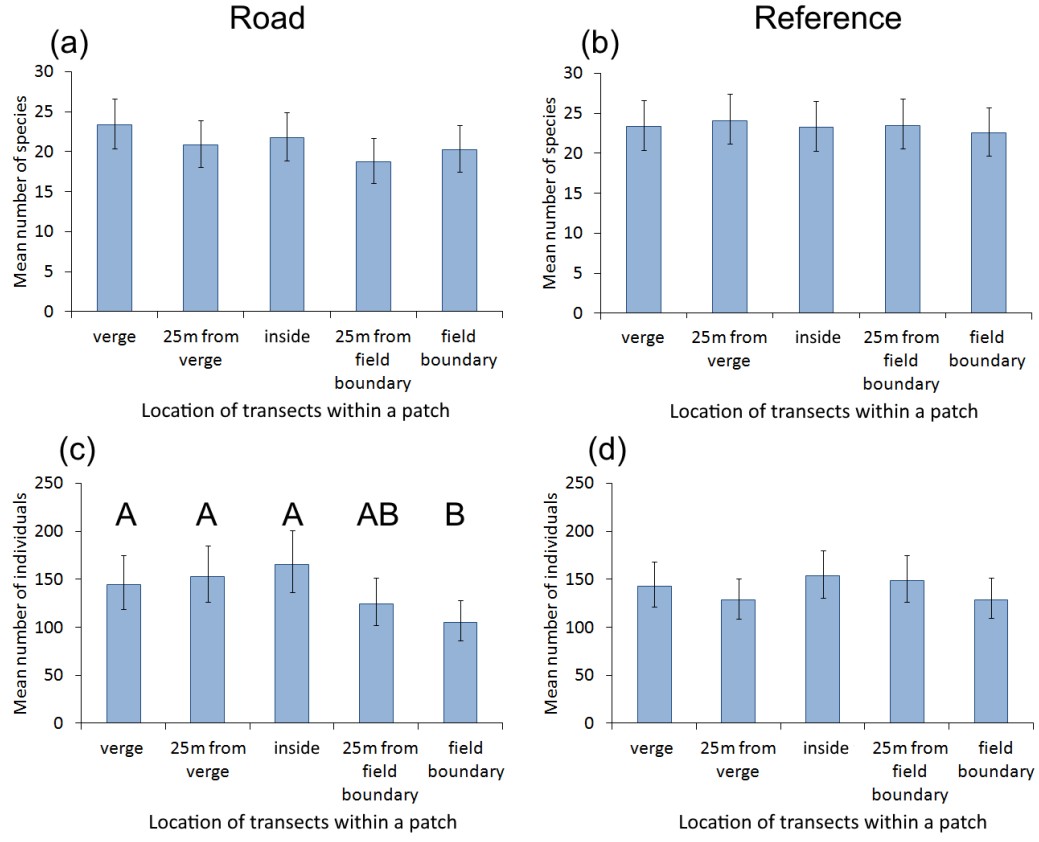

**Figure 3** **The impact of transect location within a grassland patch on the number of butterfly species and individuals.** The impact of transect location on the mean number of species (A, B) and individuals (C, D) within grassland patches adjacent to roads with traffic (A, C) and those far from roads (B, D). Whiskers are 95% confidence intervals. The only statistically significant differences were found for the abundance of butterflies (C): locations with different capital letters are significantly different.

indicated that only transects located in road verges ($F = 2.6$, $P = 0.0025$) and 25 m from road verges ($F = 1.5$, $P = 0.0187$) significantly differentiated plant communities within grassland patches adjacent to roads with traffic.

In the reference grassland patches, there were no differences in plant species richness (GLMM, $F_{4,45} = 0.274$, $P = 0.893$, Fig. S3) and mean plant species cover between transect locations (GLMM $F_{4,45} = 0.483$, $P = 0.748$, Fig. S3).

The first two axes of the partial CCA explained 5.9% of the variation in plant species composition in reference grasslands. The location of transects within the patch explained 9.7% of this variation (Fig. S4). However, no ordination axis was statistically significant (test of all axes, $F = 1.0$, $P = 0.6217$), and transect location within the grassland patch was a nonsignificant variable.

Hypothesis 4. Overall, the Co-CA indicated that the plant community (total inertia = 7.162) does not explain butterfly species composition (total inertia = 0.91): the two Co-CA axes explained 28% of the variation in butterfly community composition, but the test of the first ordination axis was statistically nonsignificant (lambda = 0.0093, $P = 0.128$), nor

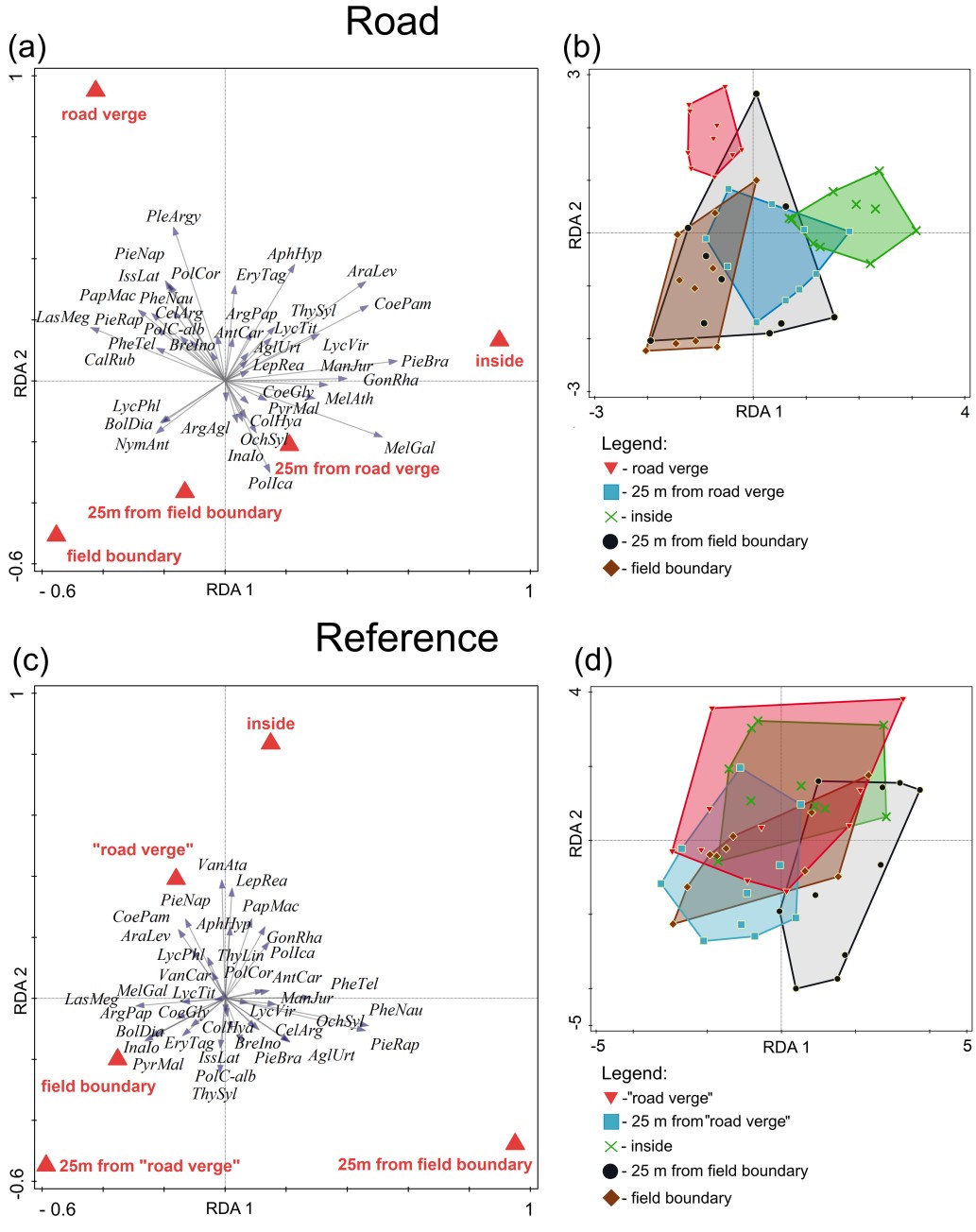

**Figure 4** **Differentiation of butterfly species in different parts of the grassland patches adjacent to roads with traffic and these far from roads.** Ordination of butterfly species in a partial redundancy analysis in different parts of the grassland patches adjacent to roads with traffic (A, B) and far from a road (C, D). Road verge – transect on a road verge, 25 m from road verge – transect located inside a grassland patch 25 m from a road verge, inside – transect located in the interior of the grassland patch, 25 m from field boundary – transect located inside the grassland patch 25 m from a border between the patch and arable field, and field boundary – transect located at the border between the grassland patch and arable field. In the case of reference grassland patches (grassland located far from a road), "road verge" was a transect located along a field road used by farmers. Species abbreviations are the first letters of the genus and species names.

was the test of both ordination axes (trace $= 0.067$, $P = 0.238$). The species richness of butterflies was correlated with plant species richness ($r = 0.234$, $P = 0.0189$, $n =$ data from all 100 transects) but not with plant cover ($r = 0.171$, $P = 0.0898$, $n = 100$). The same was true for butterfly abundance, which was correlated with plant species richness ($r = 0.347$, $P = 0.0004$, $n = 100$) but not with plant cover ($r = 0.075$, $P = 0.456$, $n = 100$).

Hierarchical partitioning showed that 61% ($F = 5.8$, $P = 0.0005$) of the variation in butterfly species composition was explained by grassland type, 37% ($F = 1.6$, $P = 0.001$) by plant data (PC1 calculated from species richness and abundance), and 2% by their joint effect ($F = 2.2$, $P = 0.0005$).

Hypothesis 5. Altogether, we recorded 154 roadkilled (2.2% of all live) butterflies in grassland patches adjacent to roads. Co-CA showed that the total community composition of alive butterflies (total inertia $= 0.1598$) accounted for 47.4% of the variation in species composition of roadkilled butterflies (total inertia $= 1.0197$). However, the first ordination axis was statistically nonsignificant (lambda $= 0.0147$, $P = 0.0839$), as were both ordination axes (trace $= 0.0478$, $P = 0.305$). We did not find any significant correlation between the number of roadkilled butterflies and the number of butterflies living in grassland patches neighboring roads ($r = 0.185$, $P = 0.608$, $n = 10$; Fig. S5).

However, in another Co-CA, we found that 48.3% of the variation in the species composition of roadkilled butterflies (total inertia $= 2.737$) was explained by the species composition of butterflies living in road verges ($n = 1{,}444$ individuals, total inertia $= 0.676$, Fig. S6). The first ordination axis was statistically significant (lambda $= 0.0636$, $P = 0.0260$) in this analysis. Additionally, there was a statistically significant correlation ($r = 0.685$, $P = 0.029$, $n = 10$) between the number of roadkilled butterflies and the number of butterflies living in the verges of grassland patches adjacent to roads with traffic (Fig. 5).

## DISCUSSION

Many studies have found a negative impact of roads on insects (see review in *Munoz, Torres & Gonzalez-Megias, 2015*). The influx of pollution may change soil properties and thus conditions for plants that are food resources for adult butterflies and their larvae (*Munguira, Garcia-Barros & Cano, 2009*; *Munguira & Thomas, 1992*), and road traffic may negatively affect population size (*Baxter-Gilbert et al., 2015*; *Munoz, Torres & Gonzalez-Megias, 2015*).

Our study revealed that roads had a subtle effect on butterfly communities. Contrary to the stated hypothesis, grassland patches neighboring roads did not have lower butterfly richness nor abundance compared to grassland patches located farther from roads. This pattern was consistent after accounting for sampling effort (number of sampled individuals), as indicated by the rarefaction analysis. This important finding may be an indication that roads with medium traffic in Poland may have no direct negative impact on butterfly populations. The difference in species richness between the two grassland types was small (in total, grassland patches adjacent to roads had four species more than grassland patches located far from roads). Butterflies interact with many other species and are plant pollinators. For example, field studies (*Bakowski & Boron, 2005*; *Bakowski, Filipiak & Friz, 2010*; *Ezzeddine & Matter, 2008*) found that one butterfly species visited 2–19 plant species (mean $\pm$ $SE = 8.5 \pm 1.6$ plant species visited, $n = 12$ butterfly species). In a book

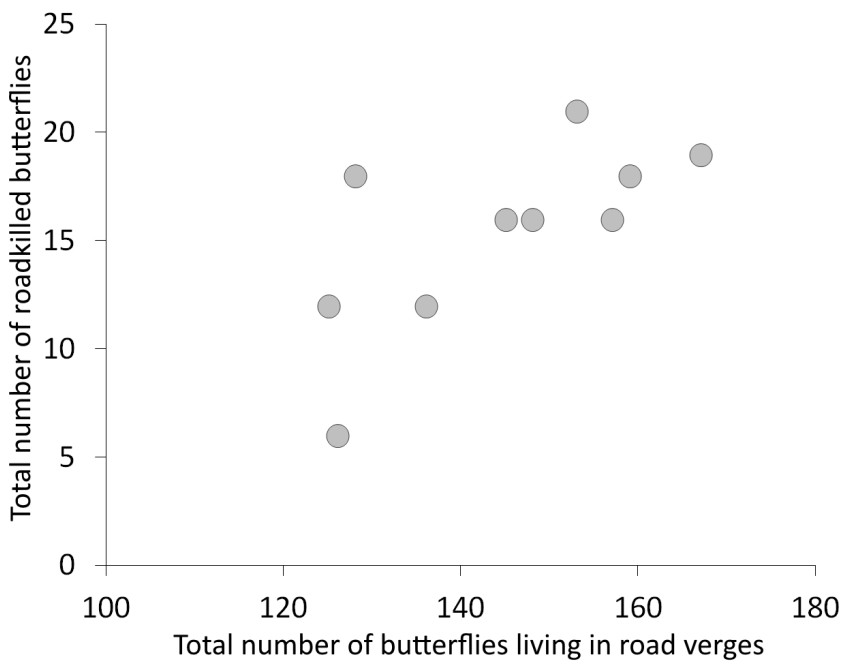

**Figure 5** Correlation between the number of butterflies living in road verges and the number of road-killed butterflies.

on British butterflies, *Dennis (2010)* reported that one butterfly species utilized 54 ± 7.5 plant species on average (range: 2–232 plant species, $n = 60$ butterfly species). Thus, each new butterfly species introduces a number of interactions among different species, and differences in butterfly species abundance between grassland types may indicate altered plant-butterfly interactions in grassland patches adjacent to roads compared to patches more distant from roads.

The explanation of the high species richness and altered species composition in grassland patches adjacent to roads is that they could be less isolated than reference grassland patches. Reference grassland patches were surrounded by arable fields, and this land cover is usually an inhospitable matrix that increases isolation (*Luoto et al., 2003*; *Lenda & Skórka, 2010*; *Öckinger et al., 2012*). As opposed to reference grasslands, patches located next to roads are connected with other grasslands adjacent to roads via road verges. These marginal habitats may improve species turnover among grasslands located near roads or even be a species pool for butterflies, enhancing patch colonization after some disturbances connected with grassland management (*Tikka, Högmander & Koski, 2001*; *Brunzel, Elligsen & Frankl, 2004*; *Moroń et al., 2017*).

In earlier works, it was stated that road verges are often good habitats for butterflies (*Ries, Debinski & Wieland, 2001*; *Wynhoff et al., 2011*; *Skórka et al., 2013a*). Road verges have specific conditions (higher influx of salt, higher temperatures, and altered water level) that may allow certain butterflies, including species of conservation concern (e.g., *Phengaris teleius* and *P. nausithous*), to survive in road verges, even in intensive agricultural

landscapes (*Wynhoff et al., 2011*). In our study, verges were an integral part of grassland patches. However, butterfly species composition and abundances in verges were different from those inside grassland patches and at the boundaries with arable fields (Hypothesis 3, Table 1). Thus, the presence of roads probably creates an environmental gradient within a grassland patch that may be preferred by different species. This possible gradient may be related to higher plant cover or other factors (soil chemistry, microclimate, etc.) at verges that were not investigated in this study. The presence of gradients of conditions may increase available niches and boost species diversity (*Amarasekare, 2003*; *Nord & Forslund, 2015*). This may also add to the explanation of the overall high butterfly species richness in grassland patches adjacent to roads.

The boundary of a habitat patch has an important effect on the species composition in the patch (*Skórka et al., 2013b*). Butterflies may respond strongly to even subtle habitat boundaries, but those responses may be modified by the edge structure and local environmental conditions (*Ries & Debinski, 2001*). In our study, both studied boundaries—with roads and with arable fields—had different species compositions and were dominated by different butterfly species, as indicated by redundancy analysis. This finding also supports Hypothesis 3 (Table 1), which states that the adjacency of a road increases the variation in butterfly species diversity in different parts of a grassland patch. It is noteworthy that in our study, the effect was visible in terms of species composition but not in terms of total species abundance and richness. Earlier theoretical and empirical works indicated that different species respond in various ways (in terms of the spatial pattern of abundance) to habitat boundaries (*Ries & Debinski, 2001*). This may be a result of different boundary and matrix types (road and arable field) and different permeabilities of these land covers (*Skórka et al., 2013b*; *Kajzer-Bonk et al., 2016*). In reference grassland patches, the species composition at the boundary with a field road was similar to the species composition at the boundary with an arable field. Field roads are very different from asphalt roads. The former are narrow, mostly covered by grass and often managed in the same way as neighboring grassland. Thus, it is possible that field roads neighboring grasslands are perceived by butterflies no differently than adjacent habitats are.

Several species showed preferences toward living in the part of grassland patches close to road verges. These are usually small-bodied species such as *Callophrys rubi*, *Plebeius argyrognomon* and *Polyommatus coridon* and very mobile, common species such as *Issoria lathonia* and *Papilio machaon*. Interestingly, road verges had high abundances of species of conservation concern such as *Phengaris nausithous*. This species is known especially for surviving at road verges in intensively managed agricultural landscapes (*Wynhoff et al., 2011*). This pattern may also result from the fact that road verges may be good habitat for ants (*Itzhak, 2008*; *Wynhoff et al., 2011*) and enhance their dispersal (*DeMers, 1993*). Ants are hosts for larvae of these small blue butterflies. However, it should be noted that these small-bodied species may be more susceptible to collision with vehicles than larger butterfly species are. Small insects may fly over the asphalt at a low altitude while crossing the road (*Soluk, Zercher & Worthington, 2011*; *Skórka et al., 2013a*) and thus may be susceptible to deadly collision with vehicles. Larger species, on the other hand, often cross roads at a

higher speed and at a higher altitude, mainly above the height of passing cars (*Skórka et al., 2013a*).

Interestingly, the overall plant species richness and composition were not affected by roads (hypotheses 1, 2 and 3, Table 1). However, the pattern of plant cover was very similar to the pattern found for the abundance of butterflies. Butterflies are herbivores, and many of them depend on specific plants during different life stages (*Kitahara, Yumoto & Kobayashi, 2008*; *Dennis, 2010*). Our co-correspondence analysis did not confirm the dependency of butterfly species composition on plants (Hypothesis 4, Table 1). This may result from the fact that adult butterflies usually utilize several plant species as nectar sources, leading to the possibility of high overlap among butterfly species using various plant species (*Dennis, 2010*). The latter possibility seems to be confirmed by the high correlation between butterfly and plant species richness. Nevertheless, the direct effect of roads on the butterfly community remained after controlling for the effects of plant species richness and abundance, and this indicates that roads also modify insect herbivore community composition in a different manner. This may be a direct effect of road mortality, alteration to species behavior near roads and changes in microclimate conditions at roads (*Jackson & Jobàggy, 2005*; *Green, Machin & Cresser, 2006*; *Skórka et al., 2013b*).

Road mortality accounted for less than 5% of all butterflies recorded in grassland patches near roads with traffic. Contrary to our expectations (Hypothesis 5, Table 1), we showed that butterflies prone to being roadkilled were the same species that were common in the verges. Among the 34 roadkilled species, two (*Phengaris teleius* and *P. nausithous*) were endangered species, protected under Polish law. However, we found only three and five individuals of *P. teleius* and *P. nausithous*, which is 23% and 14% of all individuals recorded in transects in road verges and 5% and 4% of all individuals of these species in all transects within grassland patches adjacent to roads with traffic, respectively. Even if we assume imperfect detection of roadkilled butterflies (*Skórka, 2016*), this is still a relatively low level. It would be desirable to test the effect of roads with a higher traffic level (e.g., highways) than that in this study on butterflies. Highways are being developed in Europe; thus, an assessment of their impact on butterfly communities should be addressed in further studies. However, roads with average and low traffic are the most densely distributed in Polish landscapes and possibly have the most spatially widespread environmental impact (*Kotlarek, 2007*; *Skórka et al., 2015*).

Our study has certain limitations. With only one survey during the growing season, the measurements of plant coverage are just a snapshot in time and may not be an accurate reflection of the coverage of each species across the growing season. However, it should also be noted that in our earlier studies with repeated plant sampling, we found a strong positive correlation between the number of species and coverage recorded during consecutive counts. For example, in the study by *Skórka et al. (2013a)*, the correlation coefficient between mid-May and mid-July plant surveys in road verges was $r = 0.840$ ($P < 0.001$, $n = 600$ plots) for species richness and $r = 0.494$ ($P < 0.001$, $n = 600$) for plant coverage.

In our study, we also did not consider the survival, breeding success and larval development rate of the studied butterfly species in grassland patches. This limits our inference about the lack of a negative effect of roads on butterflies because grassland patches adjacent to roads may be ecological traps. A recent study indicated that road noise imposes stressful conditions on the development of monarch butterfly (*Danaus plexippus*) larvae in road verges of highways in North America (*Davis et al., 2018*). Moreover, the influx of road salt changes the properties of host plants and thereby impacts the larval development of butterflies, but the effect may be both positive and negative (*Snell-Rood et al., 2014*). In our studied grassland patches adjacent to roads, we observed courtship, oviposition and larval foraging. As these grassland patches are permanent, it is unlikely that the butterfly populations there were sustained by only immigrants. The migration rate is usually low in butterfly populations and may increase genetic variability rather than total population size (*Nowicki & Vrabec, 2011*; *Skórka et al., 2013b*). Moreover, our anecdotal observations in the studied grassland patches in following years revealed that a high number of butterflies remained there. As the landscape compositions around all the studied grassland patches were similar, we believe that the grassland patches adjacent to roads were not ecological traps. Nevertheless, further studies should focus on comparing the estimates of survival and reproductive success of butterflies inhabiting areas close to roads to those of butterflies living in habitats not affected by roads.

## CONCLUSIONS

Our results suggest that the proximity of a road has a specific spatial effect on butterfly communities but little effect on plants. Butterfly communities were more spatially diversified within a patch and species rich in grasslands located next to roads rather than in those far from roads. Moreover, butterfly diversity was higher in grasslands neighboring roads than in grasslands far from roads. This suggests that grasslands adjacent to roads with moderate traffic may be at least as good habitats for butterflies as the grassland patches located far from roads are. Road mortality was not very high and possibly predominantly affected individuals living on grassland edges near roads, indicating that the entire patch is not equally affected by road traffic. Several butterfly species have home ranges (do not move freely throughout the entire patch); thus, these individuals occurring inside the patch may not be affected by roads even though the patch is located next to the road. Additionally, potential mitigation actions (e.g., appropriate management) may be focused on the part of the grassland located close to the road, not on the entire patch. This may minimize the costs of the mitigation actions. Such actions would enhance road verge habitats and would reduce road mortality. We propose to limit the mowing of road verges in grassland patches because it was shown in a former study (*Skórka et al., 2013a*) that limiting mowing increases the suitability of a verge as a habitat for butterflies and reduces their collisions with cars.

### Funding

This study was financed by the Polish Ministry of Science and Higher Education under project number N N304 030139. Magdalena Lenda was funded by "Mobilność Plus", a program of the Polish Ministry of Science and Higher Education and the Australian Research Council Centre of Excellence for Environmental Decisions (CE11001000104) and Australian Government. The funders had no role in study design, data collection and analysis, decision to publish, or preparation of the manuscript.

### Grant Disclosures

The following grant information was disclosed by the authors:
Polish Ministry of Science and Higher Education: N N304 030139.
Australian Research Council Centre of Excellence for Environmental Decisions: CE11001000104.
Australian Government.

### Competing Interests

The authors declare there are no competing interests.

### Author Contributions

- Piotr Skórka conceived and designed the experiments, performed the experiments, analyzed the data, contributed reagents/materials/analysis tools, prepared figures and/or tables, authored or reviewed drafts of the paper, approved the final draft.
- Magdalena Lenda and Dawid Moroń conceived and designed the experiments, performed the experiments, analyzed the data, contributed reagents/materials/analysis tools, authored or reviewed drafts of the paper, approved the final draft.

### Field Study Permissions

The following information was supplied relating to field study approvals (i.e., approving body and any reference numbers):

Field procedures were approved by the panel experts in the Ministry of Science and Higher Education (approval: 0301/B/P01/2010/39).

### Data Availability

The raw data are provided in Supplemental File.

### Supplemental Information

Supplemental information for this article can be found online at http://dx.doi.org/10.7717/peerj.5413#supplemental-information.

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
