# Peer review of "Roads affect the spatial structure of butterfly communities in grassland patches"

_PeerJ, doi:10.7717/peerj.5413_

## Round 0.1 · original submission · Major Revisions

Please address each suggestion either in the document with tracked changes or in a separate rebuttal document using line numbers. We look forward to your resubmission.

·

Basic reporting

This manuscript would have benefited greatly from a native English speaker review before submission and I suggest review by a native English speaker during revision. The text needs substantial revision to be clearly understood.

The manuscript describes some interesting work worthy of publication. The authors have demonstrated that roadside habitat in Poland provide habitat for butterflies that is as good or better than reference sites. Because of the road network, this habitat may serve not only as important habitat in and of itself, but also as a dispersal network.

However, the manuscript suffers from several issues. The link between hypotheses/question, statistical tests, and conclusions is often weak. The ms is heavy on statistical tests and weak on biological interpretation. The reason for each statistical test should be clear, and the biological interpretation of each test should be clear. The authors may want to consider removing some redundant or less informative tests to more clearly and carefully treat the remaining tests/hypotheses. The introduction and discussion are quite short as a result. Some elucidation of background and context would help the reader understand how generalizable the results are.

The figures seem to need axes labels.

Experimental design

This study is well within the aims and scope of the journal and is interesting work worthy of publication.

A figure of the transect design, or better text description, is needed. The transect placement appears to be 5 parallel transects parallel to the road, the first one 25m from the road, the next 25m further out, and so on. But, it's not completely clear.

I couldn't understand the research questions on the first read through. The knowledge gap is not stated explicitly. The research questions are very broad. The question of whether roads affect butterfly communities seems trivial, the question is what degree and how. The reader would be interested in such things.

I have some concerns about the statistical approach. For one analysis, the transects are treated as different, and for another analysis, they are lumped together. That's not necessarily wrong, but the assumptions for statistical tests need to clear. And the variety and number of multivariate tests has me concerned about spurious results. The authors should be careful to identify mechanistic or biological reasons that can explain significant results, and if the results are difficult to explain, identify the results as a hypothesis to be tested further (McGarigal et al. 2000 Multivariate Statistics for Wildlife and Ecology Research). There is little sense in the manuscript for the difference between descriptive/exploratory and inferential/confirmatory analysis and how much weight we can place on some of the results. A significant p-value doesn't guarantee truth - there are well over 20 p-values in the ms, so a few are likely to be spurious.

Validity of the findings

Good practice is followed with treatment and reference sites. Sample sizes are excellent.

Other than concerns stated above, I have concerns with overstating the importance of different butterfly species richness between road and reference sites. 38 species at reference sites vs 42 species at road sites, even if statistically significant, seems biologically very similar. I wouldn't call such communities substantially different, unless there is something important about the 4 species missing from the reference sites. However, it does seem important that roads have very similar, even almost identical, butterfly communities as reference sites, because one might hypothesize that roads would have negative effect on butterfly communities or populations. The biology of the missing species might be quite interesting too, because it could provide hypotheses as to why roads might affect some species but not other species.

The argument that the results imply that roads are good dispersal corridors needs to be made carefully. There is no data showing movement. The argument seems to be that if roads are not affecting butterfly populations at roadsides, that they are then good for dispersal because of their distribution and connectivity on the landscape.

Some discussion is needed on what these population counts represent. The reader doesn't have sense if these are butterflies passing through the sites or resident at the sites. Is the surrounding landscape habitat for these butterflies? Are the butterflies mostly confined to the sites? So do population counts likely represent a closed, resident population? Or are they a measure of butterflies that are passing through? Are all the butterflies multi-voltine and reproducing for the entire period from April to September? The reader needs some sense of the biology and landscape to be able to interpret what these counts mean.

In closing, I would like to reiterate that I find this work important and interesting and I look forward to seeing it published.

Additional comments

Please see additional comments in the attached pdf.

Reviewer 2 ·

Basic reporting

There are some grammatical inconsistencies throughout this article. I recommend that it is edited carefully to improve English grammar.

Experimental design

I think the experimental design as it pertains to the butterfly data is very novel. The research fills knowledge gaps of butterfly communities on road verges. I do think that the plant community data as some limitations which are not currently noted in the manuscript; namely only one sample was made during the growing season. In North America, one vegetation sample of cover would not adequately measure plant communities because some would have senesced by then.

Validity of the findings

The data seems sound and the benefits to the literature are clearly stated.

Additional comments

I'd like to see you address roadside management in the methods section as it pertains to the road verges in the study, and a brief overview in the discussion, since management has a huge influence on butterflies.

Thanks for your work on this study!

Annotated reviews are not available for download in order to protect the identity of reviewers who chose to remain anonymous.

---

## Round 0.2 · accepted · Accept

Thank you for your work incorporating the reviewer comments.

# ·

Basic reporting

no comment

Experimental design

no comment

Validity of the findings

no comment

Additional comments

Article is much improved and very interesting to read.